**Subject Category:**
Biology (whole organism)

behaviour/ecology

acoustic characteristics, cultural transmission, Hawaiian honeycreepers

**Author for correspondence:**
Kristina L. Paxton
e-mail: kpaxton@hawaii.edu

# Loss of cultural song diversity and the convergence of songs in a declining Hawaiian forest bird community

Kristina L. Paxton[1], Esther Sebastián-González[2], Justin M. Hite[3], Lisa H. Crampton[3], David Kuhn[4] and Patrick J. Hart[1]

[1]Department of Biology, University of Hawai'i Hilo, Hilo, HI 96720, USA
[2]Department of Applied Biology, Miguel Hernández University, Avenida de la Universidad s/n 03202 Elche, Spain
[3]Kauai Forest Bird Recovery Project, Pacific Cooperative Studies Unit, Hawaii Division of Forestry and Wildlife, University of Hawai'i Manoa, Honolulu, HI, USA
[4]SoundsHawaiian, PO Box 1018, Waimea, HI 96796, USA

KLP, 0000-0003-2321-5090; ES-G, 0000-0001-7229-1845; LHC, 0000-0002-5420-4338

The effects of population decline on culturally transmitted behaviours in animals have rarely been described, but may have major implications to population viability. Learned vocal signals in birds are of critical importance to behaviours associated with reproduction, intrasexual interactions and group cohesion, and the complexity of vocal signals such as song can serve as an honest signal of an individual's quality as well as the viability of a population. In this study, we examined how rapid population declines recently experienced by Hawaiian honeycreepers on the island of Kaua'i (USA) may have influenced the diversity, complexity and similarity of learned honeycreeper songs. We analysed the acoustic characteristics of songs recorded during three time periods over a 40-year time frame for three species of declining Kaua'i honeycreepers. We detected a loss of song complexity and diversity over the 40-year time period that paralleled dramatic population declines. Concurrent with the loss of complexity, we also found that the acoustic characteristics of the three honeycreepers' songs became more similar to one another. To our knowledge, this is the first documentation of convergence of acoustic characteristics among rapidly declining species. The reduction in song complexity and diversity and convergence of songs not only signals a loss of culturally transmitted behaviours in these endemic Hawaiian honeycreepers, but also potential challenges to the recovery of these rapidly declining species. Moreover, the present study highlights that there is a

'hidden' cost to declining populations beyond just the loss of individuals that is not often considered, the loss of culturally transmitted social behaviours.

# 1. Introduction

The effects of population decline on culturally transmitted behaviours in animals have rarely been described, yet may have major implications to behavioural ecology and population viability. In many bird (e.g. oscine passerines, hummingbirds, parrots) and mammal (e.g. cetaceans, pinnipeds, elephants, bats, primates) taxa, vocal signals are acquired from conspecifics through social learning and imitation [1–5], and are of critical importance to behaviours associated with reproduction, intrasexual interactions, individual identity and group cohesion [6,7]. Vocal signals probably facilitated the evolution of complex and dynamic social relationships found within these groups. Bird song is one of the best examples of a complex cultural trait in a non-human animal, where song complexity (e.g. number of unique song elements) in some bird species may serve as an honest signal of male quality to potential mates and competing males [8]. Increased complexity within learned songs has not only been associated with higher individual fitness in a number of bird species (e.g. reproductive success, major histocompatibility complex (MHC) diversity, parasite loads) [9–11], but also with higher population viability (e.g. Dupont's lark, *Cherosphilus duponti*) [12,13].

While selection may drive song complexity in individuals, there can also be strong selective forces for population-level stability of acoustic characteristics in songs over time. The pattern and characteristics of acoustic elements within a song or song structure has commonly been found to be stable over a period of time in both bird (e.g. yellow-naped amazon, *Amazona auropalliata*; white-crowned sparrows, *Zonotrichia leucophrys*; Galápagos medium ground finch, *Geospiza fortis*) [14–16] and whale populations (e.g. sperm whales, *Physeter microcephalus*; killer whales, *Orcinus orca*) [17,18]. Temporal stability of song structure often persists despite high rates of movement between bird populations [19,20], and even in whale populations that live in sympatry [18], suggesting that the adaptive value to conform to local song types in song learners can be a strong selective force maintaining song stability in some populations [21].

Exceptions to population-level song stability have primarily arisen when populations experience demographic changes associated with habitat loss and fragmentation [22] or sharp population declines [23], and also in response to increased ambient noise levels like anthropogenic sounds [24]. Changes or loss of locally prevalent song elements (e.g. song traditions) may be accelerated in small or severely declining populations because cultural drift (e.g. random loss of song elements), like genetic drift, is expected to act more strongly on patchy or small populations [25]. At the same time, there is a decreased likelihood of the creation of new song elements as the number of tutors for learning birds to sample from, the rate of cultural mutations (e.g. copying errors and innovation) and cultural transmission of new songs among dispersing individuals are limited in small populations [25–27]. Collectively, these factors have been shown to lead to reduced song repertoire size (a measure of song complexity) in multiple species of song learners as population size declines [13,28]. Given the vital role of song as a signal between conspecifics for both territorial defence and mate attraction [8], the loss of acoustic cultural information (e.g. locally prevalent song elements) within small populations may disrupt communication and potentially affect the persistence and viability of small populations [13,28,29].

In this study, we examined how rapid population declines experienced by native Hawaiian honeycreeper bird species on the island of Kaua'i over the last several decades may have influenced the diversity, complexity and similarity of present-day honeycreeper songs. Kaua'i honeycreepers, like all native Hawaiian forest birds, have experienced population declines and multiple extinctions over the last century as a result of threats such as habitat degradation, introduced diseases and introduced plants, birds and mammal species [30]. However, native bird communities on Kaua'i Island began to precipitously decline starting in the early 2000s [31], associated with changes in climate and disease prevalence of avian malaria, which is fatal to most Hawaiian honeycreepers [32]. Declines in Kaua'i honeycreeper populations were 1.4–5.7 times greater between 2000 and 2012 than documented in the previous 20 years [31]. Moreover, rapid range contractions in the last decade have limited most Kaua'i honeycreepers to a small, remote area encompassing only 40–60 km$^2$ of forest habitat in the Alaka'i Plateau [31]. If current rates of decline for Kaua'i honeycreepers continue, multiple extinctions are predicted in the coming decades [31]. Based on field observations of apparent increasing similarity in song among Kaua'i honeycreeper species, we hypothesized that rapid declines in the density and distribution of Kaua'i honeycreepers could have resulted in the loss of complexity and diversity within species' present-day songs [33] due to the lower number of individuals from which young birds can

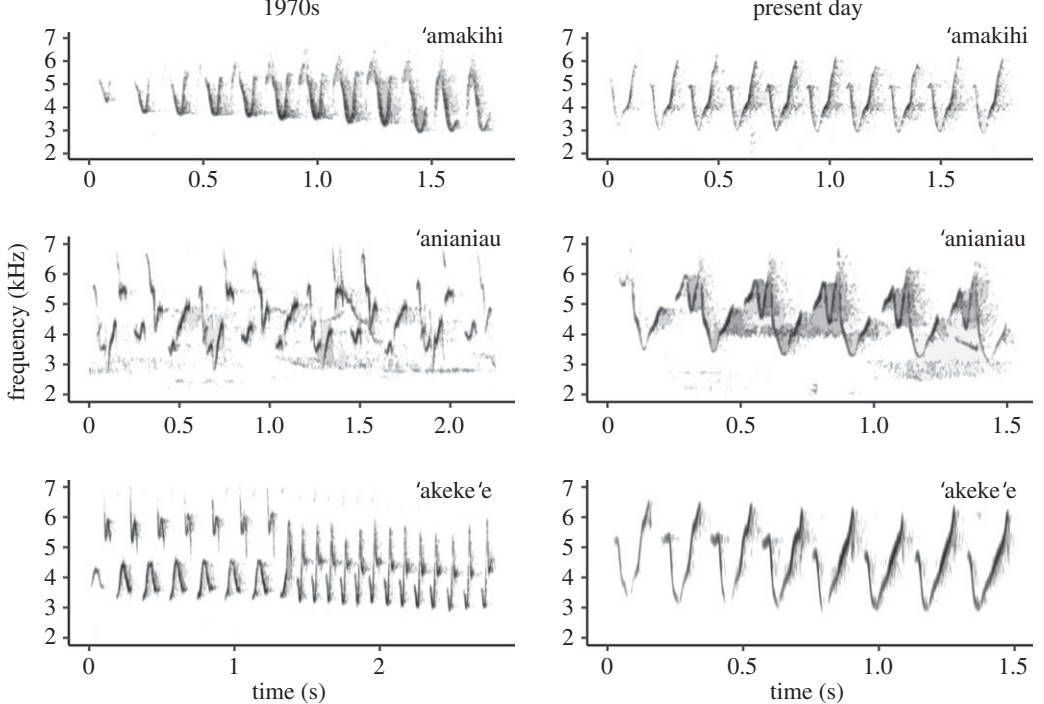

**Figure 1.** Representative songs of Kaua'i 'amakihi, 'anianiau and 'akeke'e songs from the 1970s (*a*) and present day (*b*) time periods showing loss of complexity within each species' song and the convergence of songs among species. Present-day songs have fewer syllables (e.g. Kaua'i 'amakihi, 'akeke'e) or notes (e.g. 'anianiau) and often do not include a unique beginning or ending syllable that changes in frequency from the main repeated syllable within the trill.

learn, resulting in an increase in similarity among species' present-day songs. We examined acoustic characteristics of songs recorded during three time periods (1970s, early 2000s, present day) for three species of honeycreepers: 'akeke'e (*Loxops caeruleirostris*), 'anianiau (*Magumma parva*) and Kaua'i 'amakihi (*Chlorodrepanis stejnegeri*). We specifically tested the following questions: (i) Over time has the (a) complexity and (b) variability of acoustic characteristics of Kaua'i honeycreepers' song changed within each species? (ii) Have the present-day songs of the three Kaua'i honeycreeper species become more similar to one another over time?

## 2. Methods

### 2.1. Study species and area

We examined acoustic characteristics of songs during three time periods over a 40-year time frame for three Hawaiian honeycreepers (Fringillidae) endemic to the island of Kaua'i. Male Kaua'i 'amakihi, 'anianiau and 'akeke'e all sing an undulating trill consisting of one to many unique syllables repeated throughout their song (figure 1) with low variability among the songs an individual sings. However, the unique syllables contained within a song vary among individuals within a species. All three species have ranges restricted to approximately 60 km$^2$ of forest habitat within the Alaka'i Plateau. The forest of the plateau transitions from wet montane forests dominated by 'ōhi'a (*Metrosideros polymorpha*) in the east to relatively mesic mixed 'ōhi'a-koa (*Acacia koa*) forests in the west. The current distribution of all three honeycreeper species within the Alaka'i Plateau is estimated to be only 5114–9841 ha compared with 13 200–43 400 ha in 1968 [31]. All three species have shown population declines within their core ranges over the last 25 years (i.e. 1981–2012), with 16%, 17% and 48% declines in the density of Kaua'i 'amakihi, 'anianiau and 'akeke'e, respectively [31]. However, precipitous declines in all three species' core ranges were documented between 2000 and 2012 with an average decline in abundance greater than 82% (Kaua'i 'amakihi: 91% decline; 'anianiau: 57% decline; 'akeke'e: 98% decline), and near extirpation of all three species outside of their core ranges [31]. While some declines in the density and distribution of all three species occurred during the early 2000s time period in our study, the time period between recordings in the early 2000s and present day coincides with a large acceleration in

population declines for Kaua'i 'amakihi and 'anianiau based on comparisons of density estimates from surveys conducted in 2000, 2005 and 2012 [31]. The endangered 'akeke'e has experienced the most dramatic declines with a population estimate of only 945 (95% CI: 460–1547) individuals remaining [31].

## 2.2. Acoustic recording

We obtained vocalizations of Kaua'i honeycreepers from three time periods: 1970s (1976–1978), early 2000s (2001–2004) and present day (2010–2017). Recordings from the 1970s were obtained from The Macaulay Library at the Cornell Lab of Ornithology. A single person made all recordings from the 1970s from visually identified birds using a Sony TC-45 tape recorder (Sony, USA) connected to a Sony F16 or Dan Gibson microphone in an 18 inch parabola (electronic supplementary material, table S1). Tape recordings were digitized with a sampling rate of 44.1 kHz and 24 bit WAV encoding. Recordings from the early 2000s were also made from visually identified birds by a single person using a Sony minidisc recorder (Sony, USA) connected to a Telinga PRO-8 MK2 stereo microphone (Telinga Microphones, Sweden) in a 22 inch parabola (electronic supplementary material, table S1). Present-day recordings were made using two recording set-ups: (i) Sony PCM M10 (Sony, USA) connected to a Telinga PRO-8 MK2 stereo microphone (Telinga Microphones, Sweden) in a 22 inch parabola or (ii) Marantz PMD 661 (Marantz America, LLC) connected to a Sennheiser MKH 20 microphone (Sennheiser Electronic Corporation) in a 15.5 inch parabola (electronic supplementary material, table S1). All recordings from the early 2000s and present day were recorded in a 24 bit WAV format using a 44.1 kHz sampling rate. For all three time periods, an effort was made to avoid sampling the same individual more than once by moving to a new area (e.g. greater than 25 m away) once a recording of an individual bird was completed.

Recordings from the 1970s and early 2000s were all taken within Koke'e State Park within the Alaka'i Plateau (electronic supplementary material, table S1). Given the recent reduction in honeycreeper densities with Koke'e State Park, present-day recordings were taken from both Koke'e State Park and areas along Halepa'akai Stream which is within the core range of all three species (electronic supplementary material, table S1). The maximum distance between recording locations was 10 km.

## 2.3. Acoustic analysis

Songs were spectrographically imaged and analysed using Raven Pro 1.5 software (Bioacoustics Research Program 2014) using a Hann window with a discrete Fourier transformation (DFT) size of 750 and window overlap of 50%. For each recording, we selected only high-quality songs (i.e. songs that contained minimal background noise, disturbance and did not overlap with other birds). For each song, we measured 11 acoustic characteristics (electronic supplementary material, figure S1): (i) song length (total length of song from beginning to end, in seconds), (ii) total number of syllables, (iii) number of unique syllable types, (iv) trill rate (total syllables/song length), (v) average number of notes per syllable, (vi) average number of frequency changes (directional change, either ascending or descending) within a syllable, (vii) number of frequency changes between syllables within the song, (viii) peak frequency (dominant frequency, level at which the most energy is expelled within the song, in kHz), and (ix) low (kHz) and (x) high (kHz) frequencies at which the amplitude exceeded −24 dB relative to the peak frequency [34,35]. We selected −24 dB as the threshold value because it captured variation in frequency bandwidth while excluding background noise. We also calculated (xi) frequency bandwidth (kHz) as the difference between high and low frequency. Notes were defined as a continuous vocal utterance, and a syllable defined as one or more notes grouped to form a single coherent unit that is repeated within a song. We averaged acoustic measurements for individuals with multiple high-quality songs.

## 2.4. Statistical analysis

To compare differences in acoustic characteristics of songs among time periods, we first assessed each acoustic variable for approximate normality and transformed, $\ln(x + 0.1)$, three variables to remove skewness: song length, average number of notes per syllable and average number of frequency changes within a syllable. Individually for each species, we visualized differences among time periods using a principal component analysis (PCA) based on a correlation matrix of the 11 acoustic characteristics. We retained principal component axes with eigenvalues greater than 1 and calculated an acoustic distance matrix as the Euclidean distance between pairs of individual songs defined by

the principal components scores. To test for differences in the acoustic characteristics of songs among time periods and variability within time periods, we used the R package vegan [36] to run a permutational multivariate analysis of variance (PERMANOVA; function adonis) and permutational multivariate analysis of dispersion (PERMDISP; function betadisper), respectively. PERMANOVA tests for differences in the locations (e.g. centroids) of multivariate groups [37], while PERMDISP focuses on homogeneity of multivariate dispersions (e.g. distance of observations to their centroids) [38]. When designs are balanced, PERMANOVA tests remain reliable when heterogeneity in group dispersions is present [37]. To account for large disparities in sample sizes between time periods for Kaua'i 'amakihi (electronic supplementary material, table S2), we ran PERMANOVA and PERMDISP for Kaua'i 'amakihi with both the full dataset and a reduced dataset with six randomly chosen recordings for each time period. We found no difference between the two analyses (electronic supplementary material, tables S3 and S4), and so only present the results of the full dataset. $p$-values for test statistics (pseudo-$F$) of the main effects test and posterior pairwise comparisons were based on 9999 permutations. We adjusted $p$-values of posterior pairwise comparisons to account for multiple comparisons using a Bonferroni correction and Tukey HSD for the PERMANOVA and PERMDISP analyses, respectively.

To assess if acoustic characteristics of present-day songs of the three honeycreeper species were more similar to one another than comparisons among the three honeycreepers' songs from the early 2000s or 1970s, we created a separate PCA based on a correlation matrix of all three species that included all three time periods. We then calculated the Euclidean distance between all pairwise comparisons of PCA component scores of six randomly selected recordings from each species and time period. We randomly selected six recordings to control for differences in sample size between species and time periods. We repeated this process 100 times and tested for differences between time periods by comparing the overlap in 95% CI of Euclidean distances between PCA component scores of the time periods.

To ensure changes in acoustic characteristics among time periods were a function of changes in song and not differences in the sound quality of historical tape-recorded songs and digital song recordings, we assessed potential tape degradation in historical recordings following the methods of Derryberry [39]. Historical recordings may degrade over time if they are not properly stored or digitized, given that the magnetic emulsion on audio tape is inherently unstable, and can result in frequency modulation or sound dropout from intermittent fluctuations in the tape speed. If degradation of historical recordings has occurred, we would expect more variability in the repetition of a repeated syllable in a trill in historical recordings due to sound dropout. To test this, we quantified variation in the repetition of syllables within a trill using cross-correlation spectrograms of three syllables within a single song for five individuals per species (Kaua'i 'amakihi, 'anianiau) per time period (1970s, early 2000s, present day). Cross-correlation spectrograms were calculated using the Correlator function in Raven Pro 1.5 (Bioacoustics Research Program 2014) which estimates similarity by calculating the normalized covariance between two digital spectrograms in temporal, frequency and sound energy parameters (0, no correlation; 1, exact correlation between two sounds). For each song, we calculated an average syllable similarity from the resulting cross-correlation comparisons of the three syllables within a single song. We then used a two-way ANOVA to compare the average syllable similarity within a song among time periods, species and the interaction of the two factors.

All statistical analyses were conducted using the statistical program R v. 3.4.3 (R Development Core Team 2017). Statistical significance was assumed at $\alpha = 0.05$, and standard errors are provided unless otherwise noted.

# 3. Results

We compared songs from 86 individuals, including 34 Kaua'i 'amakihi, 26 'anianiau, and 26 'akeke'e recorded during three time periods over a 40-year time frame in the Alaka'i Plateau of Kaua'i (electronic supplementary material, table S2). On average, we recorded 2.94 (±0.26) songs per individual (range 1–12 songs).

## 3.1. Kaua'i 'amakihi

We found a high degree of variability in the acoustic characteristics of Kaua'i 'amakihi songs over the 40-year time period. A PCA of the 11 measured acoustic variables showed that four principal component axes explained 78% of the variation in acoustic characteristics (electronic supplementary material, table S5). Based on a PERMANOVA of an acoustic distance matrix of the four principal component axes, we found

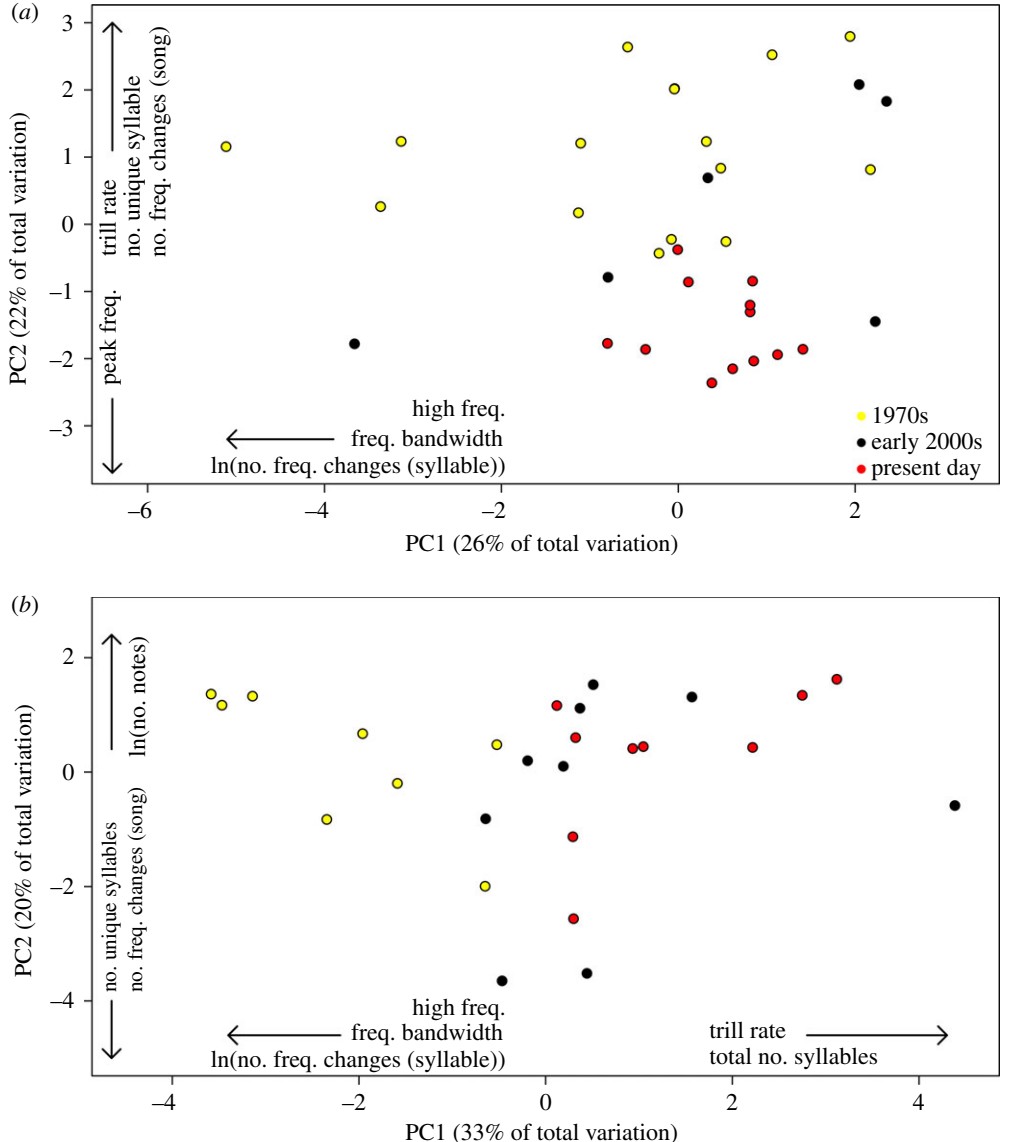

**Figure 2.** Separate PCA ordination of the standardized acoustic characteristics of each recorded song across three time periods (1970s, early 2000s, present day) for (*a*) Kauaʻi ʻamakihi and (*b*) ʻanianiau. Acoustic characteristics with the greatest PCA loadings (greater than 0.35) for each principal component axis are displayed (see electronic supplementary material, table S5 for PCA loading values).

significant differences in acoustic characteristics among time periods ($F_{2,31} = 3.96$, $p = 0.001$, $R^2 = 0.20$) (figure 2*a*). Present-day songs were significantly different compared with 1970s songs, while songs from the early 2000s were intermediate and not significantly different from either time period (table 1). Acoustic characteristics of present-day and 1970s songs differentiated the strongest along the *y*-axis of the PCA, driven by differences in variables associated with song complexity (e.g. number of unique syllables, number of frequency changes between syllables within a song). Variability along the *x*-axis of the PCA was primarily associated with frequency, and indicated that present-day songs had a narrower range of frequencies with increased low frequencies compared with songs from the 1970s and early 2000s. Moreover, songs from the 1970s and early 2000s were significantly more variable than present-day songs (PERMDISP: $F_{2,31} = 10.95$, $p < 0.001$) (table 1 and figure 2*a*).

## 3.2. ʻAnianiau

Similar to Kauaʻi ʻamakihi, ʻanianiau also had a high degree of variability in acoustic characteristics of their songs with significant differences among time periods driven by song complexity. Four PCA

**Table 1.** *Post hoc* pairwise comparisons of each time period for Kaua'i 'amakihi and 'anianiau to test for (i) differences in acoustic characteristics among time periods based on a PERMANOVA and (ii) variability in acoustic characteristics within time periods based on a PERMDISP. *p*-values of significant tests are in italics.

| comparison | differences among time periods | | variability within time periods | |
|---|---|---|---|---|
| | Kaua'i 'amakihi | 'anianiau | Kaua'i 'amakihi | 'anianiau |
| 1970s versus early 2000s | 0.99 | *0.003* | 0.52 | 0.39 |
| 1970s versus present day | *0.003* | *0.003* | *<0.001* | 0.99 |
| early 2000s versus present day | 0.29 | *0.05* | *0.001* | 0.33 |

axes with eigenvalues greater than one explained 78% of the variation in acoustic characteristics of 'anianiau songs with differences among time periods strongest along the $x$-axis (electronic supplementary material, table S5) (figure 2*b*). Acoustic characteristics of songs among all time periods were significantly different from one another with songs from early 2000s intermediate to 1970s and present-day songs (PERMANOVA: $F_{2,23} = 5.96$, $p = 0.001$, $R^2 = 0.34$) (table 1). Differences among the time periods were primarily driven by decreased complexity of 'anianiau songs over the 40-year time period, with songs recorded in the 1970s having a broader range of frequencies and more frequency changes within a syllable than present-day songs. By contrast, present-day 'anianiau songs had a faster trill rate with an increased number of times a more simple syllable was repeated (figure 2*b*). Unlike Kaua'i 'amakihi, the variability of songs within the three time periods was not significantly different among time periods (PERMDISP: $F_{2,23} = 1.32$, $p = 0.29$).

## 3.3. 'Akeke'e

Similar to patterns of the other honeycreepers, we did not find significant differences in the acoustic characteristics of 'akeke'e songs between the early 2000s and present day (PERMANOVA; $F_{2,23} = 1.38$, $p = 0.23$, $R^2 = 0.06$) (electronic supplementary material, table S5) or the variability in songs between these time periods (PERMDISP: $F_{2,23} = 1.33$, $p = 0.26$). However, we did not have enough recorded songs of 'akeke'e from the 1970s to include this time period in a statistical analysis.

## 3.4. Song similarities between species

When examining differences in the acoustic characteristics of songs among the three species, we found that there was more similarity in the acoustic characteristics of present-day songs of Kaua'i 'amakihi, 'akeke'e and 'anianiau than comparisons among the three honeycreepers' songs from the early 2000s or 1970s (figure 3). There was no overlap in the 95% confidence intervals (CI) of the average Euclidean distance between 100 randomized pairwise comparisons of PCA component scores of present-day songs ($x = 2.51$, 95% CI: 2.47, 2.54), songs recorded in the early 2000s ($x = 3.46$, 95% CI: 3.42, 3.48) or songs recorded in the 1970s ($x = 5.65$, 95% CI: 5.62, 5.69) (table 2), indicating that differences between the acoustic characteristics of the honeycreepers' present-day songs were significantly less than differences in the three honeycreepers' songs in the early 2000s and 1970s. This pattern was primarily driven by more similarity in present-day songs of Kaua'i 'amakihi with both 'akeke'e and 'anianiau (table 2).

## 3.5. Sound quality

We found no evidence to suggest that differences in acoustic characteristics of Kaua'i 'amakihi or 'anianiau songs among time periods were associated with differences in sound quality of digital and tape-recorded songs. The degradation of tape binding over time would result in a greater frequency modulation or sound dropout of historical songs compared with digital recorded songs, and thus, we would expect more variability in the highly repeatable syllable within each species' trill song. However, tape-recorded songs (i.e. 1970s) did not have greater variability in the repeated syllables within a trill song than digital song recordings (i.e. early 2000s, present day) based on comparisons of cross-correlation values among time periods and species ($F_{5,24} = 1.36$, $p = 0.27$), suggesting there was little to no degradation of sound in the historical tape-recorded songs.

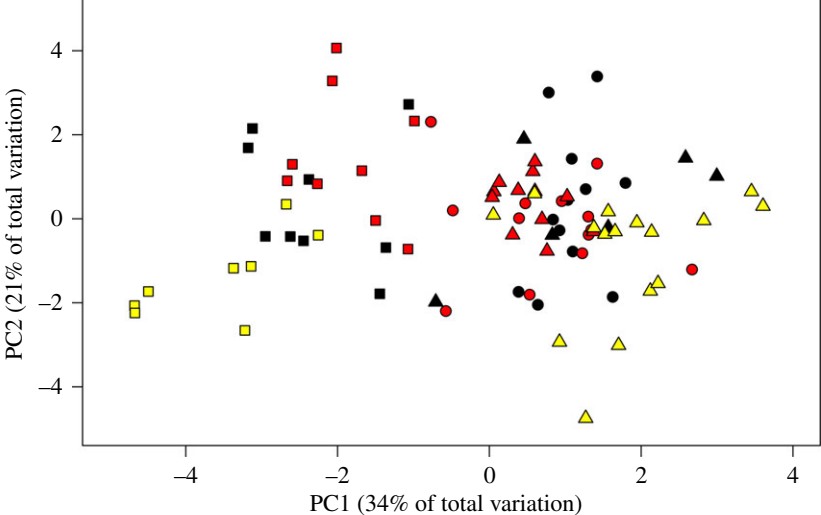

**Figure 3.** Combined PCA ordination of the standardized acoustic characteristics of all species across each time period. Colours represent different time periods (yellow, 1970s; black, early 2000s; red, present day), while shapes represent different species (triangle, Kauaʻi ʻamakihi; square, ʻanianiau; circle, ʻakekeʻe).

**Table 2.** Comparisons of the mean and 95% CI of the Euclidean distance between 100 randomized pairwise comparisons between principal component scores of Kauaʻi ʻamakihi, ʻanianiau and ʻakekeʻe songs recorded in the 1970s, early 2000s and present day. There were not enough recordings for ʻakekeʻe in the 1970s for comparisons with Kauaʻi ʻamakihi and ʻanianiau.

| comparison | 1970s | | early 2000s | | present day | |
|---|---|---|---|---|---|---|
| | mean | 95% CI | mean | 95% CI | mean | 95% CI |
| Kauaʻi ʻamakihi versus ʻanianiau | 5.65 | 5.62, 5.69 | 4.14 | 4.11, 4.17 | 3.01 | 2.97, 3.05 |
| Kauaʻi ʻamakihi versus ʻakekeʻe | — | — | 2.23 | 2.26, 2.19 | 1.49 | 1.44, 1.54 |
| ʻanianiau versus ʻakekeʻe | — | — | 3.99 | 3.95, 4.04 | 3.54 | 3.48, 3.61 |

# 4. Discussion

We detected a loss of song complexity and diversity over a 40-year time frame in multiple species of Kauaʻi honeycreepers that paralleled dramatic population declines. Concurrent with the loss of complexity in songs, we also found that the acoustic characteristics of the three honeycreepers' songs became more similar to one another. The reduction in song complexity and diversity and convergence of songs probably represents a loss of culturally transmitted behaviours in these endemic Hawaiian honeycreepers.

Over the last 25 years, Kauaʻi honeycreeper populations have rapidly declined, with an average decline in abundance greater than 82% within the three honeycreepers' core ranges in the Alakaʻi Plateau where song recordings were made [31]. The demographic changes following these steep declines most likely played a strong role in the overall loss of complexity in present-day honeycreeper songs, given the indirect role of population size on the structure and diversity of songs [25,40]. Song diversity and complexity arises through the creation of new song elements during song learning via cultural mutations (e.g. copying errors and innovation) and the cultural transmission of new songs among dispersing individuals [25]. However, based on changes in honeycreeper densities and range contractions during the course of this study, there was a two to sevenfold decrease in the density of available tutors for Kauaʻi honeycreepers to learn from, along with a 60–77% reduction in the area from which young birds could sample songs [31]. Taken together, these factors probably led to fewer adult tutors across the landscape from which juveniles could sample songs and, consequently, the reduced variability and complexity observed within present-day Kauaʻi honeycreeper songs. Consistent with these findings, small patchily distributed populations of North Island kōkako (*Callaeas wilsoni*) in New Zealand and Dupont's lark in Spain had lower song variability (e.g. increased song sharing) and smaller population song repertoire sizes, a measure of song complexity, compared with populations in larger patches [13,28,41]. However, cultural processes not

associated with demographic changes may also have contributed to the loss of complexity in Kaua'i honeycreeper songs. For example, in humpback whales (*Megaptera novaeangliae*), selection can drive the cultural evolution of songs towards more complex or simpler forms, and simple songs have been observed rapidly spreading through a population to replace a more complex song [42,43].

Interestingly, we also found that Kaua'i honeycreeper songs became more similar to one another through time. To our knowledge, this is the first documentation of convergence of acoustic characteristics among different species that are rapidly declining. This finding is consistent with recent anecdotal field observations of more overlap in Kaua'i honeycreeper songs such that distinguishing among honeycreeper songs is currently not possible without a visual conformation of a bird, but was possible some decades ago (J.M.H. and D.K. 2015, unpublished data). We posit two non-mutually exclusive hypotheses to explain acoustic convergence in honeycreepers' songs. First, Kaua'i honeycreepers' songs may have become more similar to one another as the songs of each species lost complexity over time. All three species sing a trill consisting of one to four unique syllables repeated on average nine times (range 3–24), with the acoustic structure of Kaua'i 'amakihi and 'akeke'e being the most similar (figure 1). The loss of song complexity has led to present-day honeycreeper songs containing fewer unique syllables and fewer frequency changes within and among syllables. For example, present-day songs often do not include a unique beginning or ending syllable that changes in frequency from the main repeated syllable within the trill (figure 1). Thus, random drift in the songs of all species may have led to simplified trills with similar acoustic characteristics. Alternatively, the convergence of Kaua'i honeycreeper songs may have resulted from the incorporation of song elements from other honeycreepers' songs as juveniles encountered fewer singing conspecific tutors across the landscape. Within declining populations, there are not only fewer individuals for juveniles to copy and imitate, but the vocal activity of birds can decrease as a result of reduced interactions with conspecifics [26,44], further diminishing opportunities for young birds to be exposed to a diversity of songs. In the absence of singing conspecific tutors, juveniles may acquire song elements from other nearby vocalizing species, which would result in the vocalizations of these species becoming more similar over time. While birds are predisposed to learn songs of their own species [45,46], there is some evidence that song learners in small populations (e.g. less than 10 individuals) may incorporate song elements from heterospecifics when there are few conspecifics present [12,47]. The density of honeycreeper populations has rapidly declined for all three species sampled in this study; however, current density estimates of 'akeke'e and Kaua'i 'amakihi are the lowest ('akeke'e: 0.21 birds ha$^{-1}$, Kaua'i 'amakihi: 0.61 birds ha$^{-1}$, 'anianiau: 1.66 birds ha$^{-1}$; [31]), and likewise, we saw the strongest convergence in their songs.

The loss of complexity within present-day Kaua'i honeycreeper songs may be a reason for concern in regard to population viability and persistence of Kaua'i honeycreepers. While the consequences of population declines are typically thought of in terms of the loss of genetic diversity, the disruption or loss of learned traditions can also affect species persistence, particularly when social learning is an important driver of behaviours that influence survival and reproduction [12]. The importance of cultural traditions to survival and reproduction has been demonstrated in social species where group cohesion and the transmission of social knowledge among group members are critical to the group's success. For instance, the loss of older, more experienced African elephants (*Loxodonta africana*) who model behaviour for younger individuals can negatively influence the social knowledge and reproductive success of the group as a whole [48]. In addition, it has been hypothesized that the recovery of North Atlantic right whales (*Eubalaena glacialis*) from extensive whaling is probably inhibited by the loss of knowledge of alternative feeding grounds for whales to use when conditions are poor in the Gulf of Main, the primary feeding grounds of current populations [49]. Even in less social animals that do not live in cohesive groups such as song birds, cultural traditions such as learned songs are critical to overall reproductive success, given their role in mate choice and territorial defence [50]. Sexual selection often drives complexity and repertoire size of learned songs, and consequently, in many species, these traits can be used as a reliable indicator of an individual's fitness [8]. The loss of song complexity can potentially influence effective pair formation and the efficiency of territory establishment and defence, with cascading effects on population growth. For example, small populations of Dupont's lark and North Island kōkako not only had less complex songs compared with larger populations, but populations with reduced song complexity also had lower population growth [12,28] and lower recruitment of immigrants in the case of Dupont's lark [12]. Collectively, these studies highlight that population-level assessment of song complexity has the potential to be used in monitoring population health, and that the impoverishment of present-day Kaua'i honeycreeper songs signals not only a loss of cultural traditions, but also potential challenges to

recovery of these rapidly declining species. Moreover, there is currently no indication of hybridization between Hawaiian honeycreepers [51], but we may predict that convergence in songs among Kaua'i honeycreepers could lead to a breakdown in species barriers, given the role of song in mate selection, and consequently an increase in heterospecific pairings.

Directional change in Kaua'i honeycreeper songs over time may also have resulted from selective mechanisms such as competition for acoustic space with other species [52] or the influence of habitat structure on the transmission of acoustic signals [53]. Over the course of the study, the forest structure in some areas within the Alaka'i Plateau where honeycreeper populations reside became more open as a result of hurricane damage in the 1980s and 1990s [54,55]. If adaptations to a changing acoustic environment were also driving change in the acoustic characteristics of honeycreeper songs, according to the acoustic adaptation hypothesis [56], we would expect songs from the 1970s to have lower frequencies compared with present-day songs to minimize interference with denser vegetation and maximize the transmission of song through vegetation. However, we found songs in the 1970s had a broader range of frequencies with increased high frequencies within songs compared with present-day songs, suggesting that changes in habitat structure through the course of the study were probably not driving the observed song evolution. Likewise, competition for acoustic space [57] among Kaua'i forest bird species in the Alaka'i Plateau most likely did not play a strong role in the evolution of honeycreepers' song during the course of the study, given large reductions in the densities of all Kaua'i forest birds [31], and the extinction of three species (i.e. kama'o, *Myadestes myadestinus*; Kaua'i 'o'o, *Moho braccatus*; 'o'u, *Psittirostra psittacea*) in the mid to late 1980s [58]. The lack of any stable Kaua'i honeycreeper population did not allow for the inclusion of a control species in this study to tease apart alternative hypotheses and more definitively establish the relationship between population declines and the loss of song diversity. However, we believe the effect of population size on song characteristics is the most coherent explanation of the loss of complexity within present-day Kaua'i honeycreepers' songs.

The present study highlights that there is a 'hidden' cost to declining populations beyond just the loss of individuals that is not often considered, the loss of cultural traditions. The incorporation of cultural diversity into conservation efforts is a relatively new concept [59]. However, the promotion of phenotypic diversity of learned behaviours that influence fitness parameters is essential for population resilience to environmental change, both anthropogenic and natural [60]. Kaua'i honeycreepers are precipitously declining with the potential extinction of the 'akikiki and 'akeke'e in the coming decade [31]. Captive breeding programmes have been established for these two species, and the incorporation of diverse vocalizations from which young captive birds can learn their songs will increase the likelihood of successful establishment of captive birds with wild populations in the future [12,28].

Data accessibility. All sound files used in this study have been archived at The Macaulay Library at the Cornell Lab of Ornithology, a scientific archive for research, education and conservation. Electronic supplementary material, table S1 contains a column named Record ID with the identification number given to each archived file. A table of acoustic characteristics extracted from each sound file can be accessed at the Dryad Digital Repository: https://doi.org/10.5061/dryad.t533f64 [61].

Authors' contributions. All authors contributed to the idea and design of the study; D.K., J.M.H. and K.L.P. recorded vocalizations used in the study; K.L.P. analysed the data and wrote the manuscript; P.J.H., E.S.-G., J.M.H. and L.H.C. made substantial edits to the manuscript. All authors gave final approval for publication.

Competing interests. We have no competing interests.

Funding. Funding for this research was provided by a National Science Foundation (NSF) Centers for Research Excellence in Science and Technology (CREST) grant (0833211). E.S.-G. was supported by Juan de la Cierva (MEC; IJCI-2015-24947) and by Generalitat Valenciana (SEJI/2018/024) funds. Any opinions, findings, and conclusions or recommendations expressed in this material are those of the authors and do not necessarily reflect the views of NSF.

Acknowledgements. We thank The Macaulay Library at the Cornell Lab of Ornithology for loaning acoustic recording gear to J.M.H., and Douglas H. Pratt for Kaua'i honeycreeper songs recorded in the 1970s. We also thank E. Paxton and the Listening Observatory for Hawaiian Ecosystems (LOHE) bioacoustics lab for valuable discussions on the research.

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
