## [Reviewer comments · Royal Society Open Science]

Review History

RSOS-190719.R0 (Original submission)

Review form: Reviewer 1 (Marjorie Sorensen)

Is the manuscript scientifically sound in its present form?

Yes

Are the interpretations and conclusions justified by the results?

Yes

Is the language acceptable?

Yes

Is it clear how to access all supporting data?

Yes

Do you have any ethical concerns with this paper?

No

Have you any concerns about statistical analyses in this paper?

No

Recommendation?

Accept with minor revision (please list in comments)

Comments to the Author(s)

This study is a very interesting investigation of the association between precipitous population declines and changes in song structure. The authors found that honeycreeper song complexity was lower following population declines, and that the songs of three different honeycreeper species were more similar following than prior to declines. I found the manuscript to be very well written and clear, making it a pleasure to read. I have included suggestions for improvement, outlined below.

- 1) Honeycreeper populations did not begin to decline precipitously until the early 2000s (line 676). Is there any information on % population declines during the time song recordings (from 2001-2004) were taken? I think it would help to clarify the association between population declines and your results from the three different time periods. For example, were the 2001-2004 songs sampled prior to population declines or after a significant portion of the population had already been lost?
- 2) Throughout the ms it would help to clarify that three different honeycreeper species are being considered, rather than individuals. For example, Line 691-692, "Have the present day songs of Kaua'i honeycreepers become more similar to one another compared to songs in the past?" clarify that this is a difference between species rather than individuals.
- 3) Line 698, are these variations in unique syllables fixed across species? Or is there variation between individuals?
- 4) Line 752-753: I think it would be useful here to include the mean and range for the number of songs analysed per individual. I also think that the sample sizes need to be clearer in the main ms. Currently they are found in Table S2. In addition, how were honeycreepers individually identified (caption Table S2); or is there a possibility that some individuals were recorded more than once?
- 5) Figure 1. This figure caption needs additional information. Please describe the differences in song structure observed between the 1970s and present day song (similar to what is described at Line 904-906). The acoustic characteristics measured are displayed across both the 1970s and present day song which introduces uncertainty as to whether acoustic characteristics are only specific to songs from each time period. I would suggest focusing on and identifying the differences between songs here. If need be, a separate figure with all measured acoustic characteristics displayed on one spectrogram can be included, such a figure would be well suited to the supplementary documents.
- 6) Line 809: Please clarify that 3 different time stamps were sampled over a 40 year period. Here, and throughout the ms this should be clarified since 'over a 40-year time period' suggests consistent sampling over 40 years.
- 7) Since the three honeycreeper species songs have increased in similarity, and are almost indistinguishable in some cases, is there any reason to suspect that hybridization rates between species will increase? Are these species known to hybridize? Perhaps an interesting point to include in the discussion of possible future impacts of declining song complexity.

Review form: Reviewer 2 (Timothy F. Wright)

Is the manuscript scientifically sound in its present form?

Yes

Are the interpretations and conclusions justified by the results?

Yes

Is the language acceptable?

Yes

Is it clear how to access all supporting data?

Yes

Do you have any ethical concerns with this paper?

No

Have you any concerns about statistical analyses in this paper?

No

Recommendation?

Accept with minor revision (please list in comments)

Comments to the Author(s)

This manuscript investigates changes in song over a forty-year period in three Hawaiian honeycreeper species that have experienced dramatic population declines over the same period. The authors make use of historical recordings from the 1970's and early 2000's and present day recordings they made to document changes in the acoustic characteristics and population-level diversity of songs consistent with the loss of learned variants due to cultural bottlenecks, a result that is generally predicted but has only been documented in a handful of cases. They also find the novel and intriguing result that the three species have converged on more similar songs over the same period, due perhaps to a breakdown of species barriers to learning or alternatively, as a result of similar selection pressures from changes in the habitat they all share. These results are likely to be of interest to a broad swath of scientists interested in animal communication, cultural evolution, and conservation.

Overall the study is well designed and the acoustic and statistical approaches are appropriate to the questions asked. The authors have paid due attention to potential confounds from different recording equipment from different time periods, which can be an issue when using historical samples. They have also considered potential issues posed by non-independence of pairwise comparisons and of variation in the sample size over different species and periods by avoiding some comparisons and by redoing some analyses with subsampled datasets. Thus, the interesting patterns they document seem robust. They are certain interesting as a well-documented case of loss of cultural diversity and of convergence. Most of my concerns are minor, and mostly have to do with issues of presentation or, in a few cases, interpretation:

L 639-641. Characterizing complexity of songs is tricky, as is documenting its effect on individual fitness and population viability, and I think there is a less consistent relationship between the two across the literature than this statement claims. This critique also applies to lines 939-940. See MacDougall-Shackleton 1997 and Byers and Kroodsma 2009 for reviews. Give the lack of a clear relationship between repertoire size and fitness I would question what is cause and effect in the population declines mentioned in 942-945.

L 698. It is a little ambiguous what is meant by “low variability among songs”. I think “from rendition to rendition by individuals” but it could be among individuals or even species.

L775. Bonferroni corrections can be pretty stringent, so consider whether any results that are marginal after Bonferroni corrections are still worth considering as biologically meaningful patterns.

L800-802. The ANOVA described is the one analysis in which the non-independence of pairwise comparisons may violate the assumptions of independent samples for the test employed. I suggest partial Mantel tests or some more sophisticated form of matrix comparison as an alternative.

L892-894. The result of convergence among the three species is the most novel one in the paper, so I think it is a mistake to hide the figure illustrating it in the supplement. I suggest converting figure 2 into a 4 panel figure in which the PCA's illustrating the change over time with the three species and the one showing all 4 species combined as the fourth.

L919-922. Density is certainly one factor that affects how many potential tutors a young bird encounters, but so is pattern and distance of dispersal. Any information on that parameters among the three species?

L947-949. Give the convergence in song one might eventually predict a breakdown of species barriers. Any evidence of an increase in heterospecific pairings or hybrid offspring?

Table 1. It would be better to have column headers that described the parameters themselves (e.g. mean differences, variability) rather than the statistical test used to test for effects in the parameters.

Table 2. I found this table and the 95% confidence intervals quite a bit of work to parse mentally. It think it would be much easier for most readers to absorb if it was a figure illustrating the change in mean \pm 95% CI for each of the three comparisons by the three time periods.

Fig1. I am a fan of spectrograms, so I would like to see spectrograms of all 3 species at the oldest and most recent time points, as that would give readers a concrete illustration of the acoustic changes across time both within and among the three species.

Fig 2. The legends refers to PCA loading values in Table 2, but this should be Table 5a.

Decision letter (RSOS-190719.R0)

24-Jun-2019

Dear Dr Paxton

On behalf of the Editors, I am pleased to inform you that your Manuscript RSOS-190719 entitled "Loss of cultural song diversity and the convergence of songs in a declining Hawaiian forest bird community" has been accepted for publication in Royal Society Open Science subject to minor revision in accordance with the referee suggestions. Please find the referees' comments at the end of this email.

The reviewers and handling editors have recommended publication, but also suggest some minor revisions to your manuscript. Therefore, I invite you to respond to the comments and revise your manuscript.

- Ethics statement

- Data accessibility

If you wish to submit your supporting data or code to Dryad (<http://datadryad.org/>), or modify your current submission to dryad, please use the following link:
<http://datadryad.org/submit?journalID=RSOS&manu=RSOS-190719>

- Competing interests

- Authors' contributions

- Acknowledgements

- Funding statement

Because the schedule for publication is very tight, it is a condition of publication that you submit the revised version of your manuscript before 03-Jul-2019. Please note that the revision deadline will expire at 00.00am on this date. If you do not think you will be able to meet this date please let me know immediately.

on behalf of Dr Alecia Carter (Associate Editor) and Kevin Padian (Subject Editor)
openscience@royalsociety.org

Associate Editor Comments to Author (Dr Alecia Carter):

Dear authors,

I have now received two reviews of your manuscript. Both reviewers agree that this study is interesting and well-executed, and I find myself in agreement. Both reviewers include some minor, constructive suggestions to help further improve an already well-presented study.

When addressing the reviewers' comments, I would encourage the authors to consider one further limitation and an alternative explanation of their data. In the first case, I would have found these data more convincing if a control species / population of honeycreeper (or other small passerine) was included in the analysis where there was no decline in population and / or density and the authors could show that there was no change in song complexity. It may be that some variable other than population density decline is driving song diversity loss in this genus, and such a control could have been within the authors' reach and needs to be addressed. (Please note that I understand that most, if not all, honeycreepers have declined in Kaua'i, but this control could be broader and should be mentioned or considered.)

In the second case, the lack of the above control allows alternative explanations to be considered. In particular, loss of song diversity arises through cultural processes alone, rather than demographic processes that impact on cultural processes. Importantly, increased complexity is not always the direction that cultural evolution takes. This is exemplified by humpback whale songs, where "simple" songs are documented to take over in populations, notably without any population decline.

E.g.: Eriksen, Nina, et al. "Cultural change in the songs of humpback whales (*Megaptera novaeangliae*) from Tonga." *Behaviour* (2005): 305-328.

Garland, Ellen C., et al. "Dynamic horizontal cultural transmission of humpback whale song at the ocean basin scale." *Current biology* 21.8 (2011): 687-691.

Cultural evolution is far more rapid in this species than in the honeyeaters, but this example does show that cultural change can result in reduced complexity in sexually-selected songs. Please consider this alternative explanation.

Reviewer comments to Author:

Reviewer: 1

Comments to the Author(s)

This study is a very interesting investigation of the association between precipitous population declines and changes in song structure. The authors found that honeycreeper song complexity was lower following population declines, and that the songs of three different honeycreeper species were more similar following than prior to declines. I found the manuscript to be very well written and clear, making it a pleasure to read. I have included suggestions for improvement, outlined below.

1) Honeycreeper populations did not begin to decline precipitously until the early 2000s (line 676). Is there any information on % population declines during the time song recordings (from 2001-2004) were taken? I think it would help to clarify the association between population declines and your results from the three different time periods. For example, were the 2001-2004 songs sampled prior to population declines or after a significant portion of the population had already been lost?

2) Throughout the ms it would help to clarify that three different honeycreeper species are being considered, rather than individuals. For example, Line 691-692, "Have the present day songs of Kaua'i honeycreepers become more similar to one another compared to songs in the past?" clarify that this is a difference between species rather than individuals.

3) Line 698, are these variations in unique syllables fixed across species? Or is there variation between individuals?

4) Line 752-753: I think it would be useful here to include the mean and range for the number of songs analysed per individual. I also think that the sample sizes need to be clearer in the main ms. Currently they are found in Table S2. In addition, how were honeycreepers individually identified (caption Table S2); or is there a possibility that some individuals were recorded more than once?

5) Figure 1. This figure caption needs additional information. Please describe the differences in song structure observed between the 1970s and present day song (similar to what is described at Line 904-906). The acoustic characteristics measured are displayed across both the 1970s and present day song which introduces uncertainty as to whether acoustic characteristics are only specific to songs from each time period. I would suggest focusing on and identifying the differences between songs here. If need be, a separate figure with all measured acoustic characteristics displayed on one spectrogram can be included, such a figure would be well suited to the supplementary documents.

6) Line 809: Please clarify that 3 different time stamps were sampled over a 40 year period. Here,

and throughout the ms this should be clarified since ‘over a 40-year time period’ suggests consistent sampling over 40 years.

7) Since the three honeycreeper species songs have increased in similarity, and are almost indistinguishable in some cases, is there any reason to suspect that hybridization rates between species will increase? Are these species known to hybridize? Perhaps an interesting point to include in the discussion of possible future impacts of declining song complexity.

Reviewer: 2

Comments to the Author(s)

This manuscript investigates changes in song over a forty-year period in three Hawaiian honeycreeper species that have experienced dramatic population declines over the same period. The authors make use of historical recordings from the 1970's and early 2000's and present day recordings they made to document changes in the acoustic characteristics and population-level diversity of songs consistent with the loss of learned variants due to cultural bottlenecks, a result that is generally predicted but has only been documented in a handful of cases. They also find the novel and intriguing result that the three species have converged on more similar songs over the same period, due perhaps to a breakdown of species barriers to learning or alternatively, as a result of similar selection pressures from changes in the habitat they all share. These results are likely to be of interest to a broad swath of scientists interested in animal communication, cultural evolution, and conservation.

Overall the study is well designed and the acoustic and statistical approaches are appropriate to the questions asked. The authors have paid due attention to potential confounds from different recording equipment from different time periods, which can be an issue when using historical samples. They have also considered potential issues posed by non-independence of pairwise comparisons and of variation in the sample size over different species and periods by avoiding some comparisons and by redoing some analyses with subsampled datasets. Thus, the interesting patterns they document seem robust. They are certainly interesting as a well-documented case of loss of cultural diversity and of convergence. Most of my concerns are minor, and mostly have to do with issues of presentation or, in a few cases, interpretation:

L 639-641. Characterizing complexity of songs is tricky, as is documenting its effect on individual fitness and population viability, and I think there is a less consistent relationship between the two across the literature than this statement claims. This critique also applies to lines 939-940. See MacDougall-Shackleton 1997 and Byers and Kroodsma 2009 for reviews. Give the lack of a clear relationship between repertoire size and fitness I would question what is cause and effect in the population declines mentioned in 942-945.

L 698. It is a little ambiguous what is meant by “low variability among songs”. I think “from rendition to rendition by individuals” but it could be among individuals or even species.

L775. Bonferroni corrections can be pretty stringent, so consider whether any results that are marginal after Bonferroni corrections are still worth considering as biologically meaningful patterns.

L800-802. The ANOVA described is the one analysis in which the non-independence of pairwise comparisons may violate the assumptions of independent samples for the test employed. I suggest partial Mantel tests or some more sophisticated form of matrix comparison as an alternative.

L892-894. The result of convergence among the three species is the most novel one in the paper, so I think it is a mistake to hide the figure illustrating it in the supplement. I suggest converting figure 2 into a 4 panel figure in which the PCA's illustrating the change over time with the three species and the one showing all 4 species combined as the fourth.

L919-922. Density is certainly one factor that affects how many potential tutors a young bird encounters, but so is pattern and distance of dispersal. Any information on that parameters among the three species?

L947-949. Give the convergence in song one might eventually predict a breakdown of species barriers. Any evidence of an increase in heterospecific pairings or hybrid offspring?

Table 1. It would be better to have column headers that described the parameters themselves (e.g. mean differences, variability) rather than the statistical test used to test for effects in the parameters.

Table 2. I found this table and the 95% confidence intervals quite a bit of work to parse mentally. It think it would be much easier for most readers to absorb if it was a figure illustrating the change in mean \pm 95% CI for each of the three comparisons by the three time periods.

Fig1. I am a fan of spectrograms, so I would like to see spectrograms of all 3 species at the oldest and most recent time points, as that would give readers a concrete illustration of the acoustic changes across time both within and among the three species.

Fig 2. The legends refers to PCA loading values in Table 2, but this should be Table 5a.

Author's Response to Decision Letter for (RSOS-190719.R0)

See Appendix A.

Decision letter (RSOS-190719.R1)

23-Jul-2019

Dear Dr Paxton,

I am pleased to inform you that your manuscript entitled "Loss of cultural song diversity and the convergence of songs in a declining Hawaiian forest bird community" is now accepted for publication in Royal Society Open Science.

on behalf of Dr Alecia Carter (Associate Editor) and Kevin Padian (Subject Editor)
openscience@royalsociety.org

Appendix A

UNIVERSITY
of HAWAII®
HILO

Dear Alice Power and Editors of Royal Society Open Science,

Please accept the revised manuscript (RSOS-190719) titled: "Loss of cultural song diversity and the convergence of songs in a declining Hawaiian forest bird community" for publication in Royal Society Open Science. We thank the associate editor and 2 reviewers for their thorough and insightful comments. We have responded to the minor revisions requested, which have improved our manuscript. Please see below for more detailed responses (in blue) to comments made by the associate editor and reviewers (in black). The page and line numbers indicated in our responses refer to line numbers in the revised manuscript with track changes. We hope you will find that we have addressed the comments of the reviewers and thank you for consideration of the revised manuscript for publication in Royal Society Open Science.

Sincerely,
Kristina Paxton, PhD
University of Hawaii Hilo

Response to Associate Editor

When addressing the reviewers' comments, I would encourage the authors to consider one further limitation and an alternative explanation of their data. In the first case, I would have found these data more convincing if a control species / population of honeycreeper (or other small passerine) was included in the analysis where there was no decline in population and / or density and the authors could show that there was no change in song complexity. It may be that some variable other than population density decline is driving song diversity loss in this genus, and such a control could have been within the authors' reach and needs to be addressed. (Please note that I understand that most, if not all, honeycreepers have declined in Kaua'i, but this control could be broader and should be mentioned or considered.)

In the second case, the lack of the above control allows alternative explanations to be considered. In particular, loss of song diversity arises through cultural processes alone, rather than demographic processes that impact on cultural processes. Importantly, increased complexity is not always the direction that cultural evolution takes. This is exemplified by humpback whale songs, where "simple" songs are documented to take over in populations, notably without any population decline.

E.g.: Eriksen, Nina, et al. "Cultural change in the songs of humpback whales (*Megaptera novaeangliae*) from Tonga." *Behaviour* (2005): 305-328.
Garland, Ellen C., et al. "Dynamic horizontal cultural transmission of humpback whale song at the ocean basin scale." *Current biology* 21.8 (2011): 687-691.

Cultural evolution is far more rapid in this species than in the honeyeaters, but this example does show that cultural change can result in reduced complexity in sexually-selected songs. Please consider this alternative explanation.

We agree with the associate editor that inclusion of a species that does not have population declines would allow for a more definitive conclusion regarding the relationship between population decline and song complexity. We considered this fact when designing the study, however, all honeycreeper populations on Kaua'i and even the non-native Japanese White-eyes have shown declines in the last 25 years. Non-native species that occur in the Alaka'i Plateau that have not shown population declines include the Japanese bush-warbler and Hwamei. However, given large differences in the structure of their

songs (e.g., non-trill) compared to the species included in this study, it would be hard to measure many of the acoustic characteristics used to assess change in song (e.g. frequency change within a syllable, frequency change within a song) in these non-native species. We have included text in the discussion for why a control species was not included in the study (lines 337-341). In addition, we have included text in the discussion about alternative hypotheses for the loss of complexity in song and the humpback whale examples (lines 266-269).

Response to Reviewers

Reviewer: 1

Comments to the Author

This study is a very interesting investigation of the association between precipitous population declines and changes in song structure. The authors found that honeycreeper song complexity was lower following population declines, and that the songs of three different honeycreeper species were more similar following than prior to declines. I found the manuscript to be very well written and clear, making it a pleasure to read. I have included suggestions for improvement, outlined below.

1) Honeycreeper populations did not begin to decline precipitously until the early 2000s (line 676). Is there any information on % population declines during the time song recordings (from 2001-2004) were taken? I think it would help to clarify the association between population declines and your results from the three different time periods. For example, were the 2001-2004 songs sampled prior to population declines or after a significant portion of the population had already been lost?

We have added a sentence in the methods section to better describe population changes within the time periods in the study (lines 99-102). The sentence reads - While some declines in the density and distribution of all three species occurred during the early 2000s time period in our study, the time period between recordings in the early 2000s and present day coincides with a large acceleration in population declines for Kaua'i 'amakihi and 'anianiau based on comparisons of density estimates from surveys conducted in 2000, 2005, and 2012.

2) Throughout the ms it would help to clarify that three different honeycreeper species are being considered, rather than individuals. For example, Line 691-692, "Have the present day songs of Kaua'i honeycreepers become more similar to one another compared to songs in the past?" clarify that this is a difference between species rather than individuals.

We have clarified our questions to reflect that the first question is examining differences within a species and the second question is examining differences between species (lines 81-83). We have also clarified this concept throughout the manuscript (lines 161-162, 222-230)

3) Line 698, are these variations in unique syllables fixed across species? Or is there variation between individuals?

We have clarified the text to indicate that syllables within each species' songs vary among individuals (line 89-91)

4) Line 752-753: I think it would be useful here to include the mean and range for the number of songs analysed per individual. I also think that the sample sizes need to be clearer in the main ms. Currently they are found in Table S2. In addition, how were honeycreepers individually identified (caption Table S2); or is there a possibility that some individuals were recorded more than once?

Information on the mean and range of the number of songs analyzed per individual is included in the results section (lines 194-195). We have also added the overall sample size for each species to the results section (lines 193), and kept the more detailed breakdown of sample sizes by species and time period in Table S2. We have added text to the methods section to indicate that an effort was made to

avoid sampling the song of the same male more than once by moving to a new area once a recording of an individual bird was completed (lines 117-118).

5) Figure 1. This figure caption needs additional information. Please describe the differences in song structure observed between the 1970s and present day song (similar to what is described at Line 904-906). The acoustic characteristics measured are displayed across both the 1970s and present day song which introduces uncertainty as to whether acoustic characteristics are only specific to songs from each time period. I would suggest focusing on and identifying the differences between songs here. If need be, a separate figure with all measured acoustic characteristics displayed on one spectrogram can be included, such a figure would be well suited to the supplementary documents.

As suggested, we have focused Figure 1 on differences in songs between the 1970s and present day time periods. In addition, as suggested by reviewer 2, we have included example songs from the 1970s and present day time periods for each species. We made a separate figure, included in the supplemental documents – Figure S1, that identifies all measured acoustic characteristics on a spectrogram.

6) Line 809: Please clarify that 3 different time stamps were sampled over a 40 year period. Here, and throughout the ms this should be clarified since 'over a 40-year time period' suggests consistent sampling over 40 years.

We have adjusted our wording here and throughout the manuscript to reflect that we are examining 3 different time periods over a 40-year time frame (lines 88, 194, 246)

7) Since the three honeycreeper species songs have increased in similarity, and are almost indistinguishable in some cases, is there any reason to suspect that hybridization rates between species will increase? Are these species known to hybridize? Perhaps an interesting point to include in the discussion of possible future impacts of declining song complexity.

There is no evidence of hybridization in any of the Hawaiian honeycreepers (Knowlton et al 2014). Given the idea of song convergence potentially leading to an increased chance of hybridization we have included a sentence in the discussion talking about this concept (lines 320-322).

Reviewer: 2

Comments to the Author(s)

This manuscript investigates changes in song over a forty-year period in three Hawaiian honeycreeper species that have experienced dramatic population declines over the same period. The authors make use of historical recordings from the 1970's and early 2000's and present day recordings they made to document changes in the acoustic characteristics and population-level diversity of songs consistent with the loss of learned variants due to cultural bottlenecks, a result that is generally predicted but has only been documented in a handful of cases. They also find the novel and intriguing result that the three species have converged on more similar songs over the same period, due perhaps to a breakdown of species barriers to learning or alternatively, as a result of similar selection pressures from changes in the habitat they all share. These results are likely to be of interest to a broad swath of scientists interested in animal communication, cultural evolution, and conservation.

Overall the study is well designed and the acoustic and statistical approaches are appropriate to the questions asked. The authors have paid due attention to potential confounds from different recording equipment from different time periods, which can be an issue when using historical samples. They have also considered potential issues posed by non-independence of pairwise comparisons and of variation in the sample size over different species and periods by avoiding some comparisons and by redoing some analyses with subsampled datasets. Thus, the interesting patterns they document seem robust. They are certainly interesting as a well-documented case of loss of cultural diversity and of convergence. Most of my concerns are minor, and mostly have to do with issues of presentation or, in a few cases, interpretation:

L 639-641. Characterizing complexity of songs is tricky, as is documenting its effect on individual fitness and population viability, and I think there is a less consistent relationship between the two across the literature than this statement claims. This critique also applies to lines 939-940. See MacDougall-Shackleton 1997 and Byers and Kroodsma 2009 for reviews. Given the lack of a clear relationship between repertoire size and fitness I would question what is cause and effect in the population declines mentioned in 942-945.

We have adjusted our wording in both the introduction (lines 41-46) and the discussion (lines 309-310) to reflect that not all species show a consistent relationship between song complexity and fitness.

L 698. It is a little ambiguous what is meant by “low variability among songs”. I think “from rendition to rendition by individuals” but it could be among individuals or even species.

We have clarified our wording in the text to indicate that there is low variability among the songs an individual sings (lines 90-91)

L775. Bonferroni corrections can be pretty stringent, so consider whether any results that are marginal after Bonferroni corrections are still worth considering as biologically meaningful patterns.

We assessed the results of the post-hoc comparison and there was no marginally significant results in our dataset

L800-802. The ANOVA described is the one analysis in which the non-independence of pairwise comparisons may violate the assumptions of independent samples for the test employed. I suggest partial Mantel tests or some more sophisticated form of matrix comparison as an alternative.

We averaged the pairwise comparisons of the syllables contained in a single song so that there was only one value per song used as the response variable in the two-ANOVA, and thus there is no violation of the assumption of independence. We have added text in the methods to make this more clear (lines 180-182).

L892-894. The result of convergence among the three species is the most novel one in the paper, so I think it is a mistake to hide the figure illustrating it in the supplement. I suggest converting figure 2 into a panel figure in which the PCA's illustrating the change over time with the three species and the one showing all 4 species combined as the fourth.

We have moved the figure showing the convergence in songs among the three species to the main paper, and it is now Figure 3. We included the figure as a stand-alone figure and not as a panel in Figure 2 to avoid confusion between 1) the differing colors and symbols in the legends of the two figures, and 2) the separate PCAs conducted for the individual species and all three species combined.

L919-922. Density is certainly one factor that affects how many potential tutors a young bird encounters, but so is pattern and distance of dispersal. Any information on that parameters among the three species?

We do not have any information on the pattern and distance of dispersal for the three species. We have reworded the sentence so as not to indicate that density is the only factor that can affect the probability of encountering potential tutors (lines 292-295).

L947-949. Given the convergence in song one might eventually predict a breakdown of species barriers. Any evidence of an increase in heterospecific pairings or hybrid offspring?

There is no evidence of hybridization in any of the Hawaiian honeycreepers (Knowlton et al 2014). Given the idea of song convergence potentially leading to an increased chance of hybridization we have included a sentence in the discussion talking about this concept (lines 310-312).

Table 1. It would be better to have column headers that described the parameters themselves (e.g. mean differences, variability) rather than the statistical test used to test for effects in the parameters.

As suggested, we have changed the column headers to describe the parameters tested. The columns are now: "Differences among time periods" and "Variability within time periods"

Table 2. I found this table and the 95% confidence intervals quite a bit of work to parse mentally. It think it would be much easier for most readers to absorb if it was a figure illustrating the change in mean \pm 95% CI for each of the three comparisons by the three time periods.

We have retained the quantitative data showing the convergence of songs as a Table given that we 1) moved the supplemental figure showing the convergence of songs among the three species to the main paper – Figure 3, and 2) included representative spectrograms for each species during the 1970s and present day time period showing examples of convergence in songs – Figure 1. However, if the editor feels a figure would better represent the data in Table 2, we can easily make this table into a figure.

Fig1. I am a fan of spectrograms, so I would like to see spectrograms of all 3 species at the oldest and most recent time points, as that would give readers a concrete illustration of the acoustic changes across time both within and among the three species.

See response to reviewer 1 - comment #5

Fig 2. The legends refers to PCA loading values in Table 2, but this should be Table 5a.

We have change the text in the legend for Figure 2 as suggested.